# Controlling the pressure of hydrogen-natural gas mixture in an inclined pipeline

Sarkhosh S. Chaharborj[1], Norsarahaida Amin[1,2]*

**1** Department of Mathematical Sciences, Faculty of Science, Universiti Teknologi Malaysia, UTM, Johor Bahru, Malaysia, **2** Adjunct Professor, Department of Mathematics, Fakultas Sains dan Teknologi, Universitas Airlangga, Surabaya City, East Java, Indonesia

\* norsarahaida@utm.my

**Data Availability Statement:** All relevant data are within the paper and its Supporting Information files.

**Funding:** Financial support provided by Research and Innovation University Grant Scheme, Universiti Teknologi Malaysia, is gratefully acknowledged.

## Abstract

This paper discusses the optimal control of pressure using the zero-gradient control (ZGC) approach. It is applied for the first time in the study to control the optimal pressure of hydrogen-natural gas mixture in an inclined pipeline. The solution to the flow problem is first validated with existing results using the Taylor series approximation, regression analysis and the Runge-Kutta method combined. The optimal pressure is then determined using ZGC where the optimal set points are calculated without having to solve the non-linear system of equations associated with the standard optimization problem. It is shown that the mass ratio is the more effective parameter compared to the initial pressure in controlling the maximum variation of pressure in a gas pipeline.

## Introduction

The existing natural gas pipeline networks is using by many petroleum companies to transport hydrogen and natural gas in the same pipeline to deduct transportation cost [1]. Gas pipeline networks work at high pressure and use compression stations to supply gas over long distances [2]. Gas velocity, valve closure time, and arrangement of the closing valve cause pressure change in pipelines [3]. Maximum pressure can occur during valve closure at the end of the pipeline. Therefore, Short times during valve closure are important in reducing the maximum pressure. However, because of the damage is not always visible until long after the event this transient pressure is tough to control. Therefore, controlling transient pressure in a pipeline with respect to certain efficient parameters such as initial pressure and mass ratio are very important [4, 5].

The transient flow in hydrogen-natural gas mixtures has been studied by [2, 6–9]. Isothermal flow and horizontally pipeline is assumed in these papers. However, in reality most gas pipelines are not horizontal [10] studied to transport the gas flow of hydrogen-natural gas mixture with high pressure in an inclined pipeline. Since the gas properties cannot be assumed to be constant, a one-dimensional, non-isothermal gas flow model was solved to simulate the slow and fast fluid transients, such as those typically found in high pressure gas transmission pipelines in [2].

**Competing interests:** The authors have declared that no competing interests exist.

Analysis on mass ratio and pressure variation by developing numerical models and computer algorithms in paper [11, 12]. In the paper [13], studied the control of transient gas flows in complex pipeline intersections using a linear approximation of the equations describing the physics of gas flow in pipelines and formulated the optimization problem as a mixed-integer program. A computer control algorithm has been used to study optimization of gas networks under transient conditions in the paper [14]. In the paper [15], a model is proposed to predict the decompression wave speed of high-pressure hydrogen-natural gas mixtures in pipelines. In the papers [11–15], controlling the high pressure and transient pressure have been studied using modified computer algorithms and numerical methods.

In paper [16] a method called Zero Gradient Control (ZGC) was proposed for controlling transient pressure, which is used to control the heat rejection pressure and the rotational speed of the fans of the gas cooler of R-744 refrigeration cycles nearly energy optimal. It combines ideas of Extremum Seeking Control (ESC) [17] and the online algorithms [18–20].

In the present paper, the Taylor series, regression analysis and zero gradient control have been applied to control the pressure. This is the first time, ZGC method is applied in hydrogen natural gas mixture in an inclined pipeline to control pressure. The basic idea of ZGC is to not to have to calculate set points but to use controllers bringing the gradient to zero to solve the non-linear system of equations associated with the standard optimization problem.

Remind that, solving the equations of hydrogen-natural gas mixtures in pipelines without assuming the celerity pressure wave as constant is more complicated and cannot be solved very easily [9]. To overcome these difficulties, one can use the Taylor series of celerity pressure wave. In the following, approximations for the density and celerity pressure will be proposed. The zero gradient will be applied to approximated series to control the transient pressure. We will study the effect of initial pressure and mass ratio on the maximum values of transient pressure. Effect of initial pressure and transient pressure on the optimal values of mass ratio will be considered for the control of the pressure.

## Mathematical formulation

Fig 1 describes an inclined pipeline with a reservoir at upstream and a valve downstream. The governing equations included three coupled non-linear hyperbolic partial differential

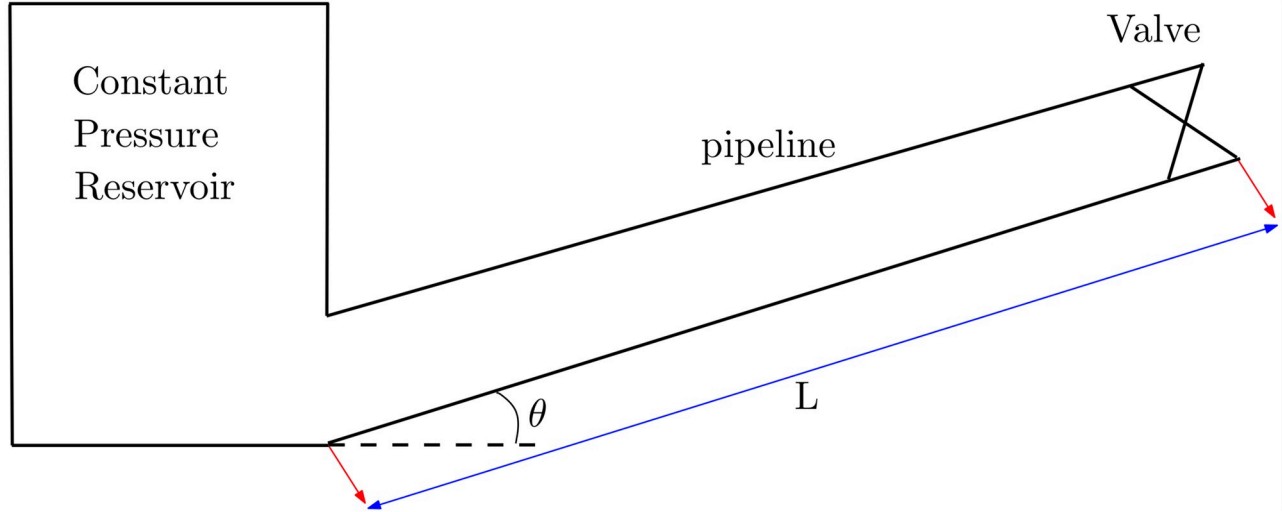

**Fig 1. The system consists of a simple inclined pipeline with a reservoir upstream and a valve downstream of the pipeline.**

equations. The flow was assumed as one dimensional, non-isothermal, compressible and covering transient condition. The fluid was assumed to be a homogeneous hydrogen and natural gas mixture [2, 6].

## Governing equations

From references [6] the principle of conservation of mass and momentum laws, the governing equations and the initial and boundary conditions for the transportation of hydrogen-natural gas mixture in an inclined pipeline and using the relation,

$$P = c^2 \rho \tag{1}$$

can be written as,

$$\frac{\partial P}{\partial t} + \frac{\partial Pu}{\partial x} = 0, \tag{2}$$

$$\frac{\partial Pu}{\partial t} + \frac{\partial (Pu^2 + c^2)}{\partial x} + \frac{fPu|u|}{2D} + Pg\sin\theta = 0, \tag{3}$$

with the boundary conditions as follows,

$$P(0,t) = P_0(t), \quad u(0,t) = u_0(t), \text{ (at the initial point, } x = 0), \tag{4}$$

$$P(L,t) = P(t), \quad u(L,t) = u_L(t), \text{ (at the initial point, } x = L), \tag{5}$$

with $u_0 = Q_0/A$ and $A = \pi D^2/4$, where $\rho$ is density, $u$ is defined as gas velocity where the modulus sign is to ensure that the frictional force shall always act opposite to the direction of motion, $P$ is the pressure, $e$ is the gas internal energy per unit mass, $f$ is the coefficient of friction, $D$ is the diameter of the pipeline, $g$ is the gravitational force and $\theta$ is an angle between the friction force and the $x$ direction.

The equation of state for perfect gas, which is commonly used in the gas industry, is given by, $P = \rho RT$, where $R$ is the specific gas constant and $T$ is the temperature. For compressible flow, the relation of equation of state with the celerity pressure wave $c$ is denoted as, $P = c^2\rho$. From relation of ideal gas, the specific heat at constant volume $C_v$, the specific heat at constant pressure $C_P$, the specific gas constant $R$, pressure $P$ and flow process index $\gamma$ are defined as,

$$C_p - C_v = R, \ \gamma = \frac{C_p}{C_v}, C_v = \frac{R}{\gamma - 1}. \tag{6}$$

## Hydrogen-natural gas mixture equation

The mass ratio and the density of hydrogen-natural gas mixture are defined as,

$$\phi = \frac{m_h}{m_h + m_g}, \quad \frac{1}{\rho} = \frac{v_h + v_g}{m_h + m_g}, \tag{7}$$

with $v_h = \frac{m_h}{\rho_h}, \ v_g = \frac{m_g}{\rho_g}, \ \rho_h = \rho_{h_0}\left(\frac{P}{P_0}\right)^{\frac{1}{n_1}}, \ \rho_g = \rho_g\left(\frac{P}{P_0}\right)^{\frac{1}{n_2}}.$

Therefore, the expression of the average density of the gas mixture is given by,

$$\rho = \left[\frac{\phi}{\rho_{h_0}}\left(\frac{P_0}{P}\right)^{\frac{1}{n_1}} + \frac{1 - \phi}{\rho_{g_0}}\left(\frac{P_0}{P}\right)^{\frac{1}{n_2}}\right]^{-1}. \tag{8}$$

**Table 1. Hydrogen properties.**

| Symbol | Fluid properties | Values (J/kgK) | |
|--------|------------------|----------------|----------------|
| | | Hydrogen | Natural gas |
| $C_p$ | Specific heat at constant pressure | 14600 | 1497.5 |
| $C_v$ | Specific heat at constant volume | 10440 | 1056.8 |
| $R$ | Gas constant | 4160 | 440.7 |

Working conditions, $P = 35$ bar and T $= 15\,^\circ C = 288$ K [6].

**Table 2. Parameters used for the simulation [6].**

| Symbol | Values | Symbol | Values |
|--------|--------|--------|--------|
| Pipe length | L = 600 m | Mass ratio | ∅ = 0, 0.5, 1 |
| Time | t = 20 s | Angle | $\theta = 0, \pi/6, \pi/4, \pi/3$ |
| Pipe diameter | D = 0.4 m | Initial mass flow | $PQ_0 = 55$ kg/s |
| Friction coefficient | F = 0.03 | Initial absolute pressure | $P_0 = 35$ br |
| Temperature | T = 15°C = 288 K | | |

The celerity pressure wave for compressible flow is defined as,

$$c = \left(\frac{\partial \rho}{\partial P}\right)_s^{-\frac{1}{2}},$$

(9)

where the subscript $s$ is defined the constant entropy condition. The derivative of Eq (8) with respect to $P$, and substituting into Eq (9), then the celerity pressure wave yields,

$$c = \left[\frac{\phi}{\rho_{h_0}}\left(\frac{P_0}{P}\right)^{\frac{1}{n_1}} + \frac{1-\phi}{\rho_{g_0}}\left(\frac{P_0}{P}\right)^{\frac{1}{n_2}}\right] \times \left[\frac{1}{P}\left[\frac{\phi}{n_1\rho_{h_0}}\left(\frac{P_0}{P}\right)^{\frac{1}{n_1}} + \frac{1-\phi}{n_2\rho_{g_0}}\left(\frac{P_0}{P}\right)^{\frac{1}{n_2}}\right]\right]^{-\frac{1}{2}}.$$

(10)

The properties of hydrogen and natural gas used in the calculations are shown in the Table 1. For the simulation, the parameters are assumed as Table 2.

## Approximate series for density and celerity

In this section, we use the Taylor series expansion as presented in the reference [21] for finding the approximate series of Eqs (8) and (10).

### Approximate series for density

Because of the singularity point $P = 0$, to find the Taylor series of Eq (8) is not possible, to overcome this problem we can use $e^P$ instead of $P$,

$$\rho(e^P) = \left[\frac{\phi}{\rho_{h_0}}\left(\frac{P_0}{e^P}\right)^{\frac{1}{n_1}} + \frac{1-\phi}{\rho_{g_0}}\left(\frac{P_0}{e^P}\right)^{\frac{1}{n_2}}\right]^{-1}.$$

(11)

then the Taylor series of Eq (11) around point $P = 0$ from order 2 is given as follows,

$$\rho(e^P) \simeq \left[\rho(e^P)\right]_{P=0} + \frac{P}{1!}\left[\rho(e^P)\right]'_{P=0} + \frac{P^2}{2!}\left[\rho(e^P)\right]''_{P=0},$$

(12)

after replacing the derivatives $[\rho(e^P))]'_{P=0}$ and $[\rho(e^P))]''_{P=0}$ we have

$$\rho(e^P) \simeq \frac{1}{A_3}\left[1 - \frac{A_4}{A_6}\rho_{h_0}\rho_{g_0}P - \frac{\rho_{h_0}\rho_{g_0}}{n_1 n_2 A_6^2}[n_1 n_2 A_5 A_6 + A_4 A_7]P^2\right], \tag{13}$$

with $A_1, A_2, \ldots, A_7$ as defined as follows

$$A_1 = e^{\frac{\ln(P_0)}{n_1}},\ A_2 = e^{\frac{\ln(P_0)}{n_2}}\ A_3 = \frac{\phi}{\rho_{h_0}}A_1 + \frac{1-\phi}{\rho_{g_0}}A_2,\ A_4 = -\frac{\phi}{n_1\rho_{h_0}}A_1 - \frac{1-\phi}{n_2\rho_{g_0}}A_2,$$

$$A_5 = \frac{\phi}{2n_1^2\rho_{h_0}}A_1 + \frac{1-\phi}{2n_2^2\rho_{g_0}}A_2,\ A_6 = \phi\rho_{g_0}A_1 + (1-\phi)\rho_{h_0}A_2, \tag{14}$$

$$A_7 = n_2\phi\rho_{g_0}A_1 + n_1(1-\phi)\rho_{h_0}A_2$$

now the function $\rho(e^P)$ is converted to function $\rho(P)$, in Eq (15) by replacing $P$ with $\ln(P)$,

$$\rho(e^P) \simeq \frac{1}{A_3}\left[1 - \frac{A_4}{A_6}\rho_{h_0}\rho_{g_0}[\ln P] - \frac{\rho_{h_0}\rho_{g_0}}{n_1 n_2 A_6^2}[n_1 n_2 A_5 A_6 + A_4 A_7][\ln P]^2\right]. \tag{15}$$

Fig 2 shows a comparison of the density evolution with pressure between exact solution (8) and approximation series (15), for different values of the hydrogen mass fraction $\phi$, by assuming an initial pressure $P_0 = 35$ bar and $T_0 = 15\,°C = 288\,K$.

## Approximate series for celerity pressure wave

To find the Taylor series of Eq (10) we follow the presented method in the previous Sections,

$$c(e^P) \simeq \left[\frac{\phi}{\rho_{h_0}}\left(\frac{P_0}{e^P}\right)^{\frac{1}{n_1}} + \frac{1-\phi}{\rho_{g_0}}\left(\frac{P_0}{e^P}\right)^{\frac{1}{n_2}}\right] \times \left[\frac{1}{P}\left[\frac{\phi}{n_1\rho_{h_0}}\left(\frac{P_0}{e^P}\right)^{\frac{1}{n_1}} + \frac{1-\phi}{n_2\rho_{g_0}}\left(\frac{P_0}{e^P}\right)^{\frac{1}{n_2}}\right]\right]^{-\frac{1}{2}}. \tag{16}$$

then the Taylor series of Eq (16) around point $P = 0$ from order 2 is given as follows,

$$c(e^P) \simeq [c(e^P)]_{P=0} + \frac{P}{1!}[c(e^P)]'_{P=0} + \frac{P^2}{2!}[c(e^P)]''_{P=0}, \tag{17}$$

after replacing the derivatives $[c(e^P))]'_{P=0}$ and $[c(e^P))]''_{P=0}$ we have

$$c(e^P) \simeq \left[\frac{\rho_{h_0}\rho_{g_0}[C_1\phi n_2\rho_{g_0} - C_2\phi n_1\rho_{h_0} + C_2 n_1\rho_{h_0}]}{n_1 n_2[C_1\phi\rho_{g_0} - C_2\phi\rho_{h_0} + C_2\rho_{h_0}]^2}\right]^{-\frac{1}{2}} - \frac{C_8 C_{12}P}{2C_6 C_7 n_1\sqrt{n_2}} + \frac{C_{13}P^2}{8C_6 C_7^2\sqrt{C_8}n_1^2 n_2^2}, \tag{18}$$

with $C_1, C_2, \ldots, C_{13}$ as defined as follows

$$C_1 = P_0^{\frac{1}{n_1}},\ C_2 = P_0^{\frac{1}{n_2}},\ C_3 = P_0^{\frac{n_2+2n_1}{n_1 n_2}},\ C_4 = P_0^{\frac{n_2+n_1}{n_1 n_2}},\ C_5 = P_0^{\frac{2n_2+n_1}{n_1 n_2}},$$

$$C_6 = \phi\rho_{g_0}C_1 + (1-\phi)\rho_{h_0}C_2,\ C_7 = n_2\phi\rho_{g_0}C_1 + n_1(1-\phi)\rho_{h_0}C_2,$$

$$C_8 = -\frac{\rho_{g_0}\rho_{h_0}C_7}{n_1 n_2 C_6^2},\ C_9 = [1-n_2]n_1^2 + [n_2^2 - 4n_2]n_1 + n_2^2,$$

$$C_{10} = [2n_1^2 + 2n_1 - 2]n_2^3 + [n_1^3 - 10n_1^2 + 6n_1]n_2^2 + [2n_1^3 - 2n_1^2]n_2 + n_1^3,$$

$$C_{11} = \frac{1}{2}[n_1+1]^2 n_2^3 + [n_1^3 - 5n_1^2 - n_1]n_2^2 + n_1^2 n_2[n_1+3] - n_1^3,$$

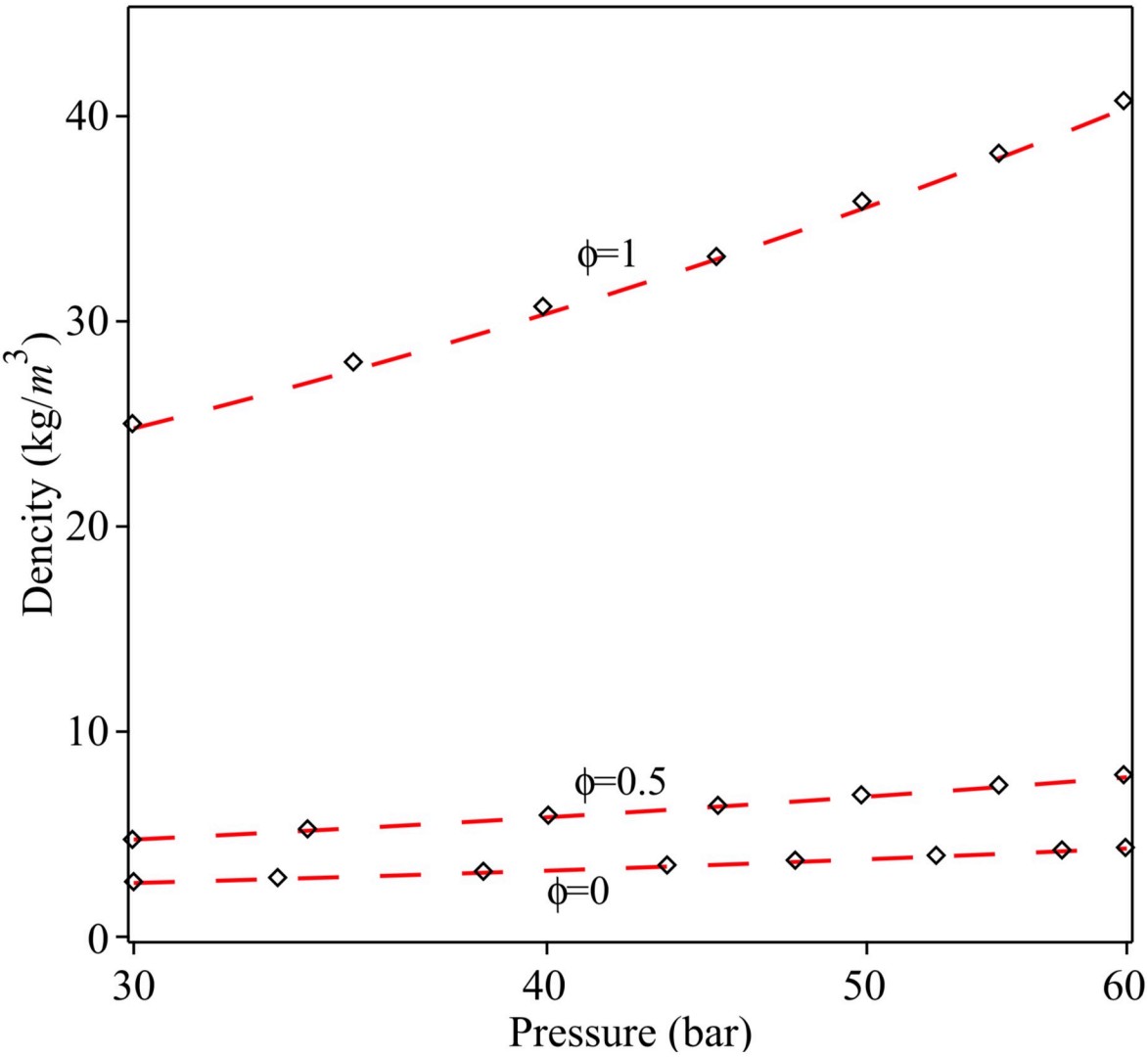

**Fig 2. Density as a function of the pressure for different values of $\phi$.** Red dash line: Proposed series and black diamond: Elaoud et al., 2010.

$$C_{12} = C_4 C_9 \rho_{g_0} \rho_{h_0} \phi (1 - \phi) + n_1^2 C_2^2 (n_2 - 1) \rho_{h_0}^2 (1 - \phi)^2 + (n_1 - 1) n_2^2 C_1^2 \rho_{g_0}^2 \phi^2,$$

$$C_{13} = C_3 C_{10} n_1 \rho_{g_0} \rho_{h_0}^2 \phi (1 - \phi)^2 - 2 C_5 C_{11} n_2 \rho_{g_0}^2 \rho_{h_0} \phi^2 (1 - \phi) + (n_1 - 1)^2 n_2^4 C_1^3 \rho_{g_0}^3 \phi^3$$
$$- (n_2 - 1)^2 n_1^4 C_2^3 \rho_{h_0}^3 (1 - \phi)^3.$$

in Eq (18) the function $\rho(e^P)$ can be converted to function $\rho(P)$, by replacing $P$ with $\ln(P)$,

$$c(e^P) \simeq \left[ \frac{\rho_{h_0} \rho_{g_0} [C_1 \phi n_2 \rho_{g_0} - C_2 \phi n_1 \rho_{h_0} + C_2 n_1 \rho_{h_0}]}{n_1 n_2 [C_1 \phi \rho_{g_0} - C_2 \phi \rho_{h_0} + C_2 \rho_{h_0}]^2} \right]^{-\frac{1}{2}} - \frac{C_8 C_{12} \ln(P)}{2 C_6 C_7 n_1 \sqrt{n_2}} + \frac{C_{13} [\ln(P)]^2}{8 C_6 C_7^2 \sqrt{C_8} n_1^2 n_2^2} \quad (19)$$

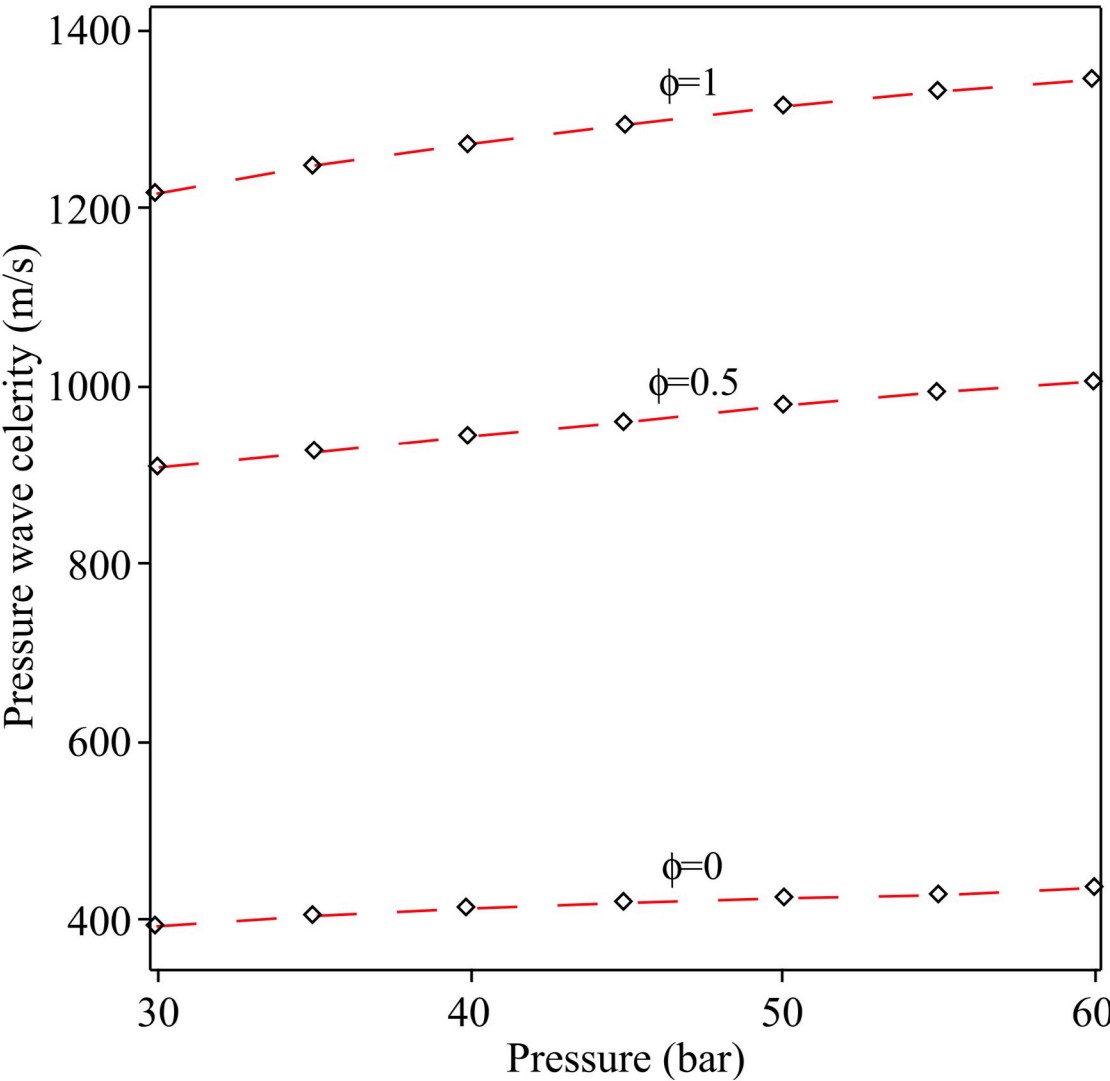

**Fig 3. Celerity pressure wave as a function of the pressure for different values of $\phi$.** Red dash line: Proposed series and black diamond: Elaoud et al., 2010.

Fig 3 shows a comparison of the density evolution with pressure between exact solution (10) and approximation series (19) for different values of the hydrogen mass fraction $\phi$ by assuming an initial pressure $P_0 = 35$ bar and $T_0 = 15$ ˚C = 288 K.

## Solution of hydrogen-natural gas mixture equations

In this section, we use the Runge-Kutta method [22] to solve the steady state equations. Regression analysis will be applied to find the regression polynomials for pressure $P$ and velocity $u$ from the numerical results. Regression polynomials can be used to solve hydrogen-natural gas mixture equations.

## Solving steady state equations

Steady state equations can be obtained from Eqs (2) and (3) for $t = 0$ as follows,

$$\frac{d(Pu)}{dx} = 0, \tag{20}$$

$$\frac{d(Pu^2)}{dx} + c^2 \frac{dP}{dx} + \frac{fPu|u|}{2D} + Pg \sin \theta = 0, \tag{21}$$

with the boundary conditions

$$u(0,0) = u(0), \; p(0,0) = P(0), \tag{22}$$

The Runge-Kutta method can be used to solve Eqs (20) and (21) as numerically. Next we use the regression analysis to find the regression polynomials for pressure $P$ and velocity $u$ as

$$P(x) = a_0 + a_1 x + a_2 x^2, \tag{23}$$

$$u(x) = b_0 + b_1 x + b_2 x^2, \tag{24}$$

The polynomials $P(x)$ and $u(x)$ will be used as initial conditions for solving Eqs (2) and (3). The unknown parameters $a_0$, $a_1$, $a_2$ and $b_0$, $b_1$, $b_2$ are found using the error function defined as

$$E = \sum_{i=0}^{m} [f(x_i) - y_i]^2, \tag{25}$$

where, $f(x_i) = P(x_i)$ or $f(x_i) = u(x_i)$ and $m$ is the number of partitions for the pipeline $L$. Here the values $x_i$ are the pipeline partition points and $y_i = y_{P,i}$ (pressure) or $y_i = y_{u,i}$ (celerity) are generated data from solving Eqs (20) and (21). With replacing the polynomials $P(x_i)$ and $u(x_i)$ in the error function (24) we have

$$E_P = \sum_{i=0}^{m} [a_0 + a_1 x + a_2 x^2 - y_{P,i}]^2, \tag{26}$$

$$E_u = \sum_{i=0}^{m} [b_0 + b_1 x + b_2 x^2 - y_{u,i}]^2. \tag{27}$$

We minimize the error functions $E_P$ and $E_u$ to find the unknown parameters $a_0$, $a_1$, $a_2$ and $b_0$, $b_1$, $b_2$ for different values of mass ration $\phi = 0, 0.5, 1$ (see Table 3).

**Table 3. Values of coefficients.**

| Parameters | $\phi = 0$ | $\phi = 0.5$ | $\phi = 01$ |
|---|---|---|---|
| $a_0$ | 34.99999945 | 34.99992670 | 34.99947059 |
| $a_1$ | -0.00043023 | -0.00215230 | -0.004032469 |
| $a_2$ | 0.0000000043023 | 0.0000000908154 | 0.0000003404764 |
| $b_0$ | 143.23945768 | 143.24069650 | 143.24866196 |
| $b_1$ | 0.00176059 | 0.00878971 | 0.01636548 |
| $b_2$ | 0.0000000363624 | 0.000000989597 | 0.0000038470726 |

Coefficients of $a_0$, $a_1$, $a_2$ and $b_0$, $b_1$, $b_2$ for various $\phi$ in Eqs (23) and (24).

We have used Maple code to find the regression polynomials as follows,

$$sol := dsolve(\{eq[1],\ eq[2],\ eq[3],\ ini\},\ numeric);$$

$$dX := [seq(x, x = 0L, h)];$$

$$P1 := [seq(abs(rhs(sol(x)[2])), x = 0L, h)];$$

$$u1 := [seq(abs(rhs(sol(x)[3])), x = 0L, h)];$$

$$P := Fit(a_0 + a_1 x + a_2 x^2, dX, P1, x);$$

$$u := Fit(b_0 + b_1 x + b_2 x^2, dX, u1, x);$$

with $eq[1]$ = Eq (20), $eq[2]$ = Eq (21), $ini$ = Eq (22), $L$ is pipe length and $h$ is pipe length partition size.

Fig 4 shows a validation between the regression polynomials of pressure $P(x)$ (Eq (23)) with the numerical results of Elaoud et al. and Subani et al. as presented in the references [7, 23] for different values of mass ratio $\phi$ = 0, 0.5, 1.

## Solving Eqs (2) and (3)

Now, by using the approximation series (19) and regression polynomials (23) and (24) with coefficients from Table 3, we can solve the Eqs (2) and (3) by the following Maple command,

$$pdsolve(\{eqs\},\ \{ini\},\ numeric,\ time = t,\ range = 0L,\ timestep = dt);$$

where, $eqs$ = {Eqs (2) and (3)}, $ini$ = {Eqs (4) and (5)} and $L$ is pipeline length.

Fig 5a, 5b and 5c shows the numerical results for the transient pressure distribution after rapid closure of the downstream valve as a function of time and for different values of the hydrogen mass ratio $\phi$. The results are in good agreement with those of reduced order modelling (ROM) presented by Subani et al. in the paper [7] and the method of characteristics (MOC) presented by Elaoud at al. in the paper [23].

The transient pressure distribution for the rapid closure valve at the downstream end of the pipe for different values of $\theta$ = 0, $\pi/6$, $\pi/4$, $\pi/3$ and mass ratio $\phi$ = 0.5 is presented in Fig 6a. The minimum and maximum transient pressures will happen in the values of $\phi$ = 0 and $\theta$ = $\pi/3$, respectively. With increasing the mass ratio from 0 to 1, transient pressure is increasing.

Fig 6b indicates the transient pressure distribution for the rapid closure valve at the downstream end of the pipe for different values of mass ratio $\phi$ = 0, 0.5, 1 and $\theta$ = 0. As can be seen, transient pressure peaks for $\phi$ = 0, $\phi$ = 0.5 and $\phi$ = 1 are 6, 4 and 2 respectively. With increasing the mass ratio from 0 to 1, transient pressure is increasing.

Fig 6c shows the transient pressure distribution for the rapid closure valve at the downstream end of the pipe for different values of permanent pressure $P_0$ = 30, 60 and $\theta$ = 0, $\phi$ = 0.5. The variation of permanent pressure $P_0$ dose not have much influence on the transient behavior.

## Zero gradient control for controlling the pressure

To find the optimum values of pressure $P$, we need to bring the gradient of Eq (8) to zero with respect to the pressure $P$. But using the basic equation of $\rho$ as proposed in the Eq (8) cannot guarantee the proper optimization for the transient pressure (see reference [24]). To overcome

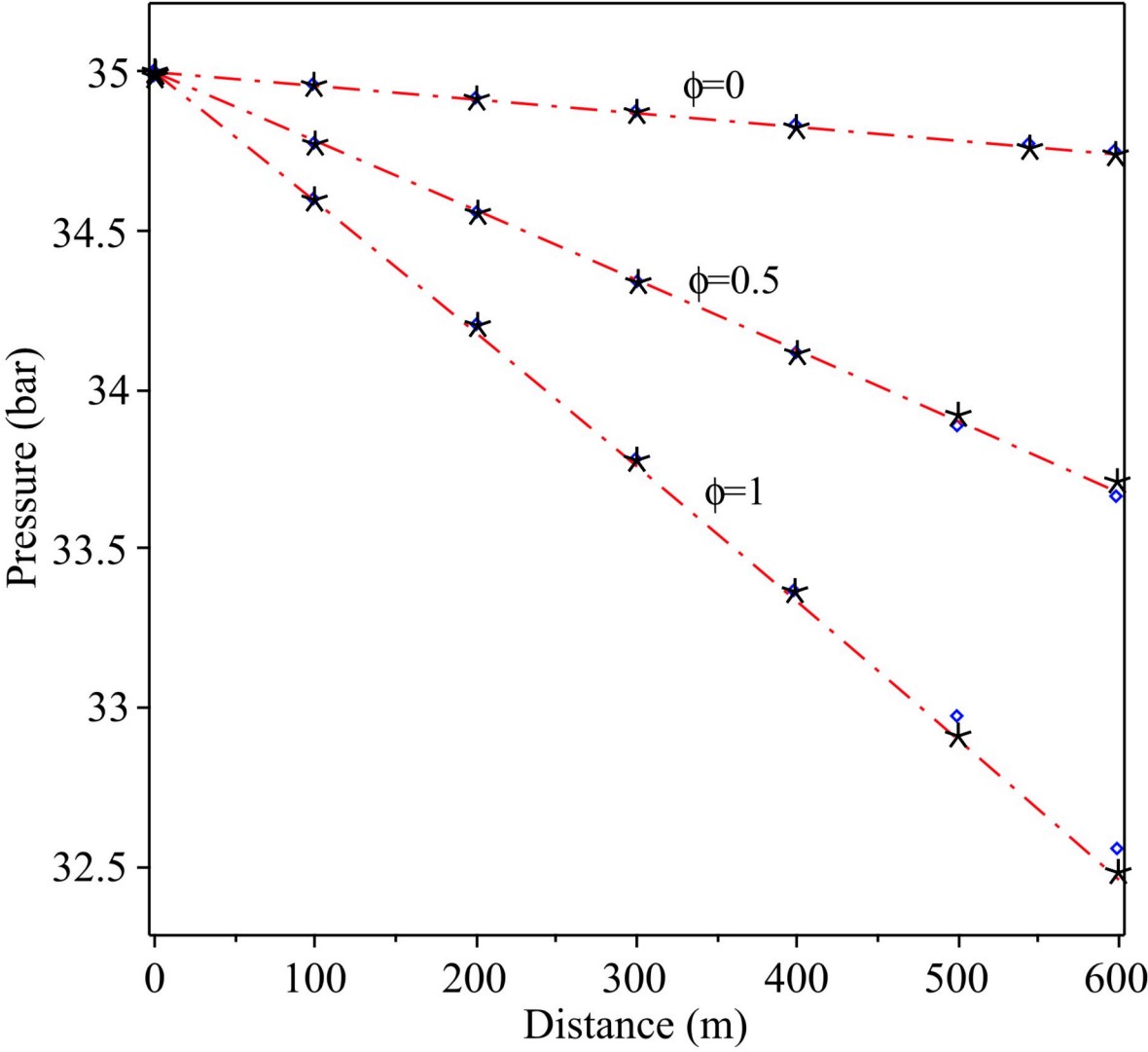

**Fig 4. Permanent regime pressure distribution along the pipe (Eqs (20) and (21)) for different mass ratio $\phi$ = 0, 0.5, 1.** Red dash dot line: Proposed series, blue diamond: Elaoud et al., 2010 and black asterisk: Subani et al., 2017.

this problem, we will use the Taylor series as shown in the Eq (15). To find the optimal values of the function $\rho(P)$ with respect to the transient pressure $P$ (optimal values of pressure), we look for the zeros gradient of the series (15) as follows,

$$\frac{\partial \rho}{\partial P} = \frac{\rho_{h_0}\rho_{g_0}[[A_2^2 n_1^2(\phi-1)^2\rho_{h_0}^2 + B_3 + A_1^2\phi^2\rho_{g_0}^2 n_2^2]\ln(P) - n_1 n_2 B_1 B_2]}{n_1^2 n_2^2 B_1^3 P}, \tag{28}$$

where $B_1$, $B_2$, $B_3$ are,

$$B_1 = \phi A_1 \rho_{g_0} - A_2(\phi - 1)\rho_{h_0},$$

$$B_2 = A_2 n_1(\phi - 1)\rho_{h_0} - \phi A_1 \rho_{g_0} n_2,$$

$$B_3 = A_1 A_2 \phi(\phi - 1)\rho_{h_0}\rho_{g_0}[n_1^2 - 4n_1 n_2 + n_2^2],$$

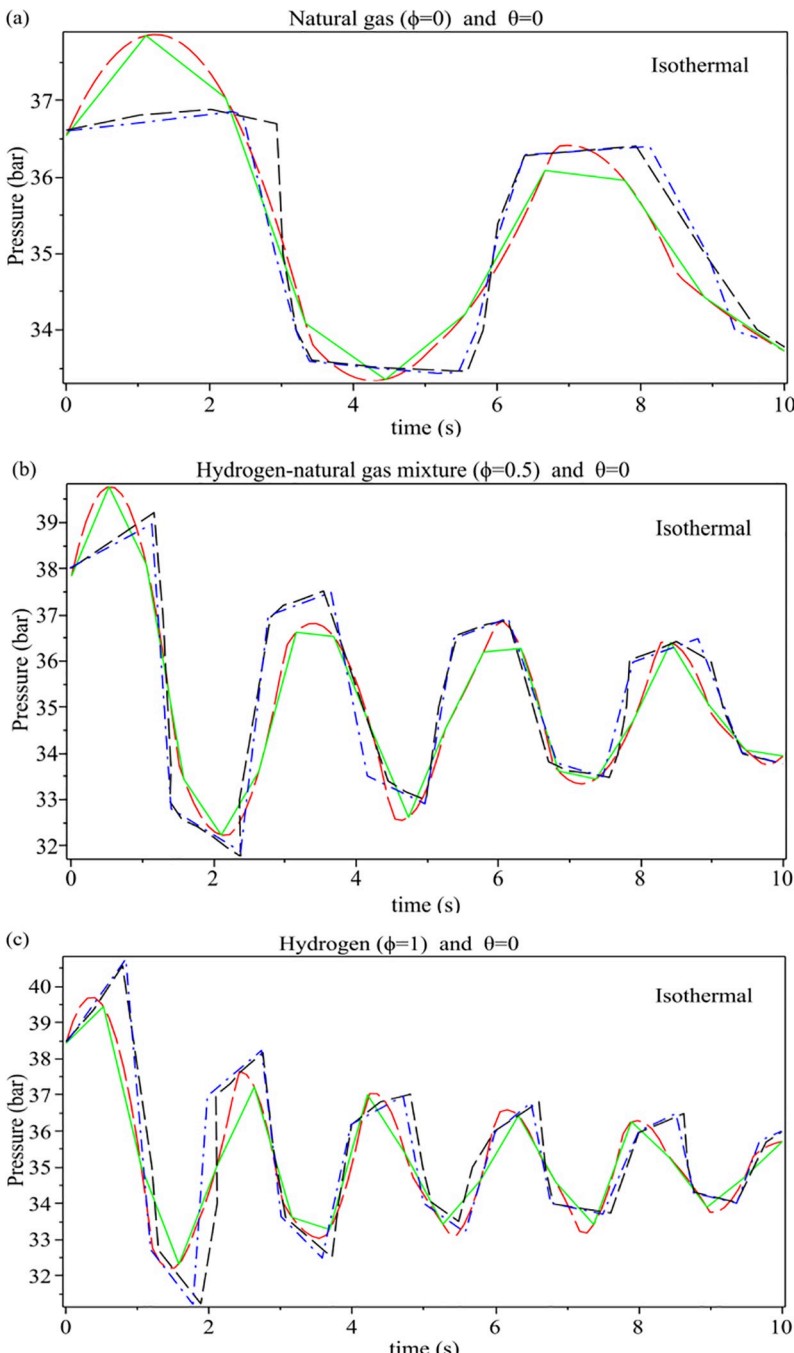

**Fig 5. Transient pressure distribution for the rapid closure valve at the downstream end of the pipe for different values of mass ratio $\phi$ and $\theta = 0$ [7, 23].** Red long dash line: Proposed method (NP = 100), green solid line: Proposed method (NP = 10), black dash line: Elaoud et al., 2010 and blue dash line: Subani et al., 2015.

solving equation $\frac{\partial \rho}{\partial P} = 0$ we can obtain the explicit formula for the optimal values of transient pressure $P$,

$$P_{\text{optimal},\rho} \simeq e^{\frac{n_1 n_2 B_1 B_2}{A_1{}^2 \phi^2 \rho_{g_0}{}^2 n_2{}^2 + \phi^2 A_2{}^2 n_1{}^2 \rho_{h_0}{}^2 - 2\phi A_2{}^2 n_1{}^2 \rho_{h_0}{}^2 + A_2{}^2 n_1{}^2 \rho_{h_0}{}^2 + B_3}}, \tag{29}$$

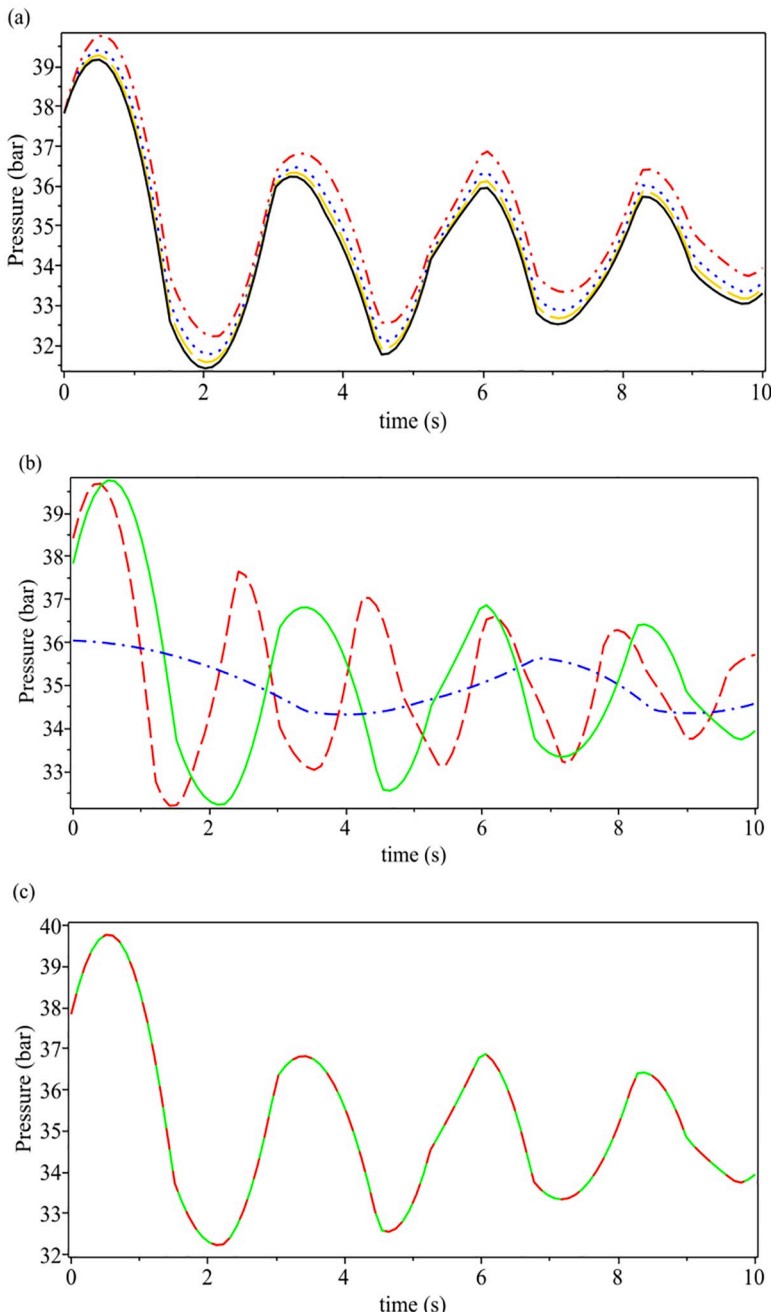

**Fig 6. Transient pressure distribution for the rapid closure valve at the downstream end of the pipe. (a): for different values of $\theta = 0$, $\pi/6$, $\pi/4$, $\pi/3$ and mass ratio $\phi = 0.5$; (b): for different values of mass ratio $\phi = 0, 0.5, 1$ and $\theta = 0$; (c): for different values of permanent pressure $P_0 = 30, 60$ and $\theta = 0$, $\phi = 0.5$.** (a): black solid line: $\theta = 0$, gold long dash line: $\theta = \pi/6$, blue dot line: $\theta = \pi/4$, red dash dot line: $\theta = \pi/3$; (b): blue dash dot line: $\theta = 0$, green solid line: $\theta = 0.5$, red dash line: $\theta = 1$; (c): green solid line: $P_0 = 30$, red long dash line: $P_0 = 60$.

Now, to find the optimum values of $c(P)$ with respect to pressure $P$ we look for the zeros gradient of the series (18), and by solving equation $\frac{\partial c}{\partial P} = 0$ we obtain the explicit formula for the optimal values of transient pressure as follows,

$$P_{\text{optimal, c}} \simeq e^{\frac{2n_1 n_2 C_7 C_{12}}{C_{13}}}, \tag{30}$$

with $C_7$, $C_{12}$ and $C_{13}$ are defined as follows,

$$C_7 = n_1(\phi - 1)\rho_{h_0}C_2 - \phi\rho_{g_0}n_2C_1,$$

$$C_{12} = C_4C_9\rho_{g_0}\rho_{h_0}\phi(1-\phi) + n_1{}^2C_2{}^2(n_2-1)\rho_{h_0}{}^2(1-\phi)^2 + (n_1-1)n_2{}^2C_1{}^2\rho_{g_0}{}^2\phi^2,$$

$$C_{13} = C_3C_{10}n_1\rho_{g_0}\rho_{h_0}{}^2\phi(1-\phi)^2 - 2C_5C_{11}n_2\rho_{g_0}{}^2\rho_{h_0}\phi^2(1-\phi) + (n_1-1)^2n_2{}^4C_1{}^3\rho_{g_0}{}^3\phi^3$$
$$-(n_2-1)^2n_1{}^4C_2{}^3\rho_{h_0}{}^3(1-\phi)^3.$$

Fig 7a and 7b show the optimum values of the pressure P as a function of the initial pressure $P_0$ and as a function of the mass ratio $\phi$, respectively; by using the Eq (29) for = 0.5 and $P_0 = 35$. As seen in Fig 7a, the optimum values of the pressure $P$ decrease $\phi$ from 0.00091710 to 0.00091700 with increasing the initial pressure $P_0$ from 30 to 60.

The results of Fig 7b show that with increasing mass ratio $\phi$ from 0 to 1, the optimal values of the transient pressure $P$ decrease from 0.00110 to 0.00090. Therefore, the variation of optimum pressure $(\nabla P_{\text{optimal}})$ with respect to the parameters initial pressure $P_0$ and mass ratio $\phi$ respectively are, $\nabla P_{\text{optimal}}(P_0) = 1 \times 10^{-7}$ and $\nabla P_{\text{optimal}}(\phi) = 2 \times 10^{-4}$. The values of the optimal pressure variations show that sensitivity of the optimal values [25–27] of the transient pressure $P$ with respect to the mass ratio $\phi$ is $\frac{\nabla P_{\text{optimal}}(\phi)}{\nabla P_{\text{optimal}}(P_0)} = \frac{2 \times 10^{-4}}{1 \times 10^{-7}} = 2000$ times more than the sensitivity with respect to the initial pressure $P_0$ (see Fig 6b and 6c).

Fig 8a and 8b shows the optimal values of the pressure $P$ as a function of the initial pressure $P_0$ and as a function of the mass ratio $\phi$, respectively; by using the Eq (29) for $\phi = 0.5$ and $P_0 = 35$. As seen in Fig 8a, the optimal values of the pressure $P$ increases from 0.2465425 (bar) to 0.2465450 (bar) with increasing initial pressure $P_0$ from 30 to 60. The results from Fig 8b shows that with increasing the mass ratio $\phi$ from 0 to 1, the optimum values of the pressure $P$ increases from 0.2425 to 0.2470. For Fig 8a and 8b we have, $\nabla P_{\text{optimal}}(P_0) = 2.5 \times 10^{-6}$, $\nabla P_{\text{optimal}}(\phi) = 4.5 \times 10^{-3}$ and $\frac{\nabla P_{\text{optimal}}(\phi)}{\nabla P_{\text{optimal}}(P_0)} = \frac{2 \times 10^{-3}}{1 \times 10^{-6}} = 1800$.

According to Figs 7a and 8b, ratio of the minimum and maximum optimal pressures is, $\frac{\text{optimal}\{P\}_{P_0=30}}{\text{optimal}\{P\}_{P_0=60}} \simeq 1.011$ and $\frac{\text{optimal}\{P\}_{P_0=30}}{\text{optimal}\{P\}_{P_0=60}} \simeq 0.99$, respectively. This means that the initial value

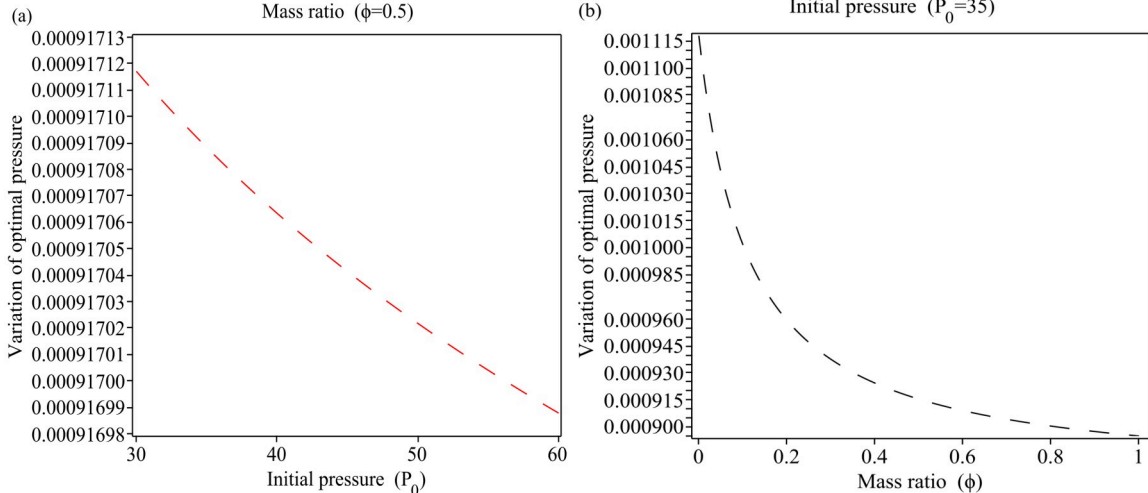

**Fig 7. Variation of optimal pressure using Eq (29).** (a): as a function of the initial pressure $P_0$, (b): as a function of the mass ratio $\phi$.

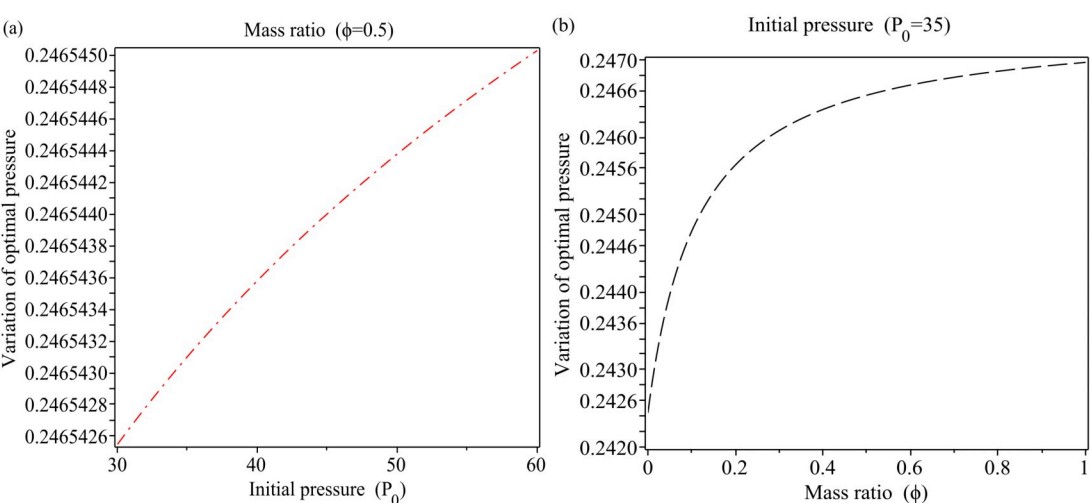

**Fig 8. Variation of optimal pressure using Eq (30).** (a): as a function of the initial pressure $P_0$, (b): as a function of the mass ratio $\phi$.

barely influences the pressure level in the pipeline. Figs 7b and 8b show that $\phi = 0.4$ is a critical point for the variation of optimum pressure and for $\phi \in [0, 0.4]$ the variation of optimal pressure is faster than $\phi \in [0.4, 1]$. We have $\frac{\text{optimal}\{P\}_{\phi=0}}{\text{optimal}\{P\}_{\phi=0.4}} \simeq 1.19$, $\frac{\text{optimal}\{P\}_{\phi=0.4}}{\text{optimal}\{P\}_{\phi=1}} \simeq 1.02$, $\frac{\text{optimal}\{P\}_{\phi=0}}{\text{optimal}\{P\}_{\phi=1}} \simeq 1.22$ and $\frac{\text{optimal}\{P\}_{\phi=0}}{\text{optimal}\{P\}_{\phi=0.4}} \simeq 0.9949$, $\frac{\text{optimal}\{P\}_{\phi=0.4}}{\text{optimal}\{P\}_{\phi=1}} \simeq 0.9943$. This means that the mass ratio is very important parameter for controlling the optimum pressure in the pipeline (see Fig 10). The Figs 7(a), 7(b), 8(a) and 8(b) show that the mass ratio is important case for controlling the optimum pressure in the pipeline. Fig 6b and 6c show the validation of the results showed in the Figs 7 and 8.

In statistics, a confidence interval (CI) is a type of interval estimate, computed from the statistics of the observed data, that might contain the true value of an unknown parameter [28–30]. Therefore, the confidence interval for the optimum values of the pressure $P$ is defined as $[\min_{P_0, \phi} P_{\text{optimal}}, \max_{P_0, \phi} P_{\text{optimal}})$, where $\min_{P_0, \phi} P_{\text{optimal}}$ and $\max_{P_0, \phi} P_{\text{optimal}}$ are given by,

$$\min_{P_0, \phi} P_{\text{optimal}} = \min_{P_0, \phi}\left\{\min_{P_0, \phi}\{\text{optimal } P_\rho\}, \ \min_{P_0, \phi}\{\text{optimal } P_c\}\right\}, \tag{31}$$

$$\max_{P_0, \phi} P_{\text{optimal}} = \max_{P_0, \phi}\left\{\max_{P_0, \phi}\{\text{optimal } P_\rho\}, \ \max_{P_0, \phi}\{\text{optimal } P_c\}\right\}, \tag{32}$$

with $\max_{P_0}[(P_{\text{optimal}})_c] = 0.246545$, $\min_{P_0}[(P_{\text{optimal}})_\rho] = 0.0009170$ and $\max_\phi[(P_{\text{optimal}})_c] = 0.2470$, $\min_\phi[(P_{\text{optimal}})_\rho] = 0.00090$.

## Pressure control by reining mass ratio

Taylor series and Zero Gradient Control are used here to control the mass ratio for obtaining efficient transient pressure. The optimum values for the mass ratio $\phi$ cannot be obtained using Eqs (8) and (10). To overcome this problem, we use the Taylor series of those equations with respect to the parameter $\phi$ of order $n$ as follows (see references [24, 30]),

$$\rho(\phi) = \rho(0) + \frac{\phi}{1!}[\rho(\phi)]'_{\phi=0} + \frac{\phi^2}{2!}[\rho(\phi)]''_{\phi=0} + \cdots + \frac{\phi^n}{n!}[\rho(\phi)]^{(n)}_{\phi=0} + \cdots, \tag{33}$$

$$c(\phi) = c(0) + \frac{\phi}{1!}[c(\phi)]'_{\phi=0} + \frac{\phi^2}{2!}[c(\phi)]''_{\phi=0} + \cdots + \frac{\phi^n}{n!}[c(\phi)]^{(n)}_{\phi=0} + \cdots, \tag{34}$$

for $n = 2$,

$$\rho(\phi) = \frac{\rho_{g_0}}{L_2} - \frac{\rho_{g_0}{}^2}{L_2{}^2}L_3\phi - \frac{\rho_{g_0}{}^2(-L_1\rho_{g_0} + L_2\rho_{h_0})}{L_2{}^3\rho_{h_0}}L_3\phi^2, \tag{35}$$

$$c(\phi) = \frac{1}{\sqrt{\frac{H_1\rho_{g_0}}{n_2 P}}} - \frac{1}{2}\frac{(H_1 n_1\rho_{h_0} - 2H_2 n_1\rho_{g_0} + H_2 n_2\rho_{g_0})}{H_1 n_1\rho_{h_0}} \cdot \frac{1}{\sqrt{\frac{H_1\rho_{g_0}}{n_2 P}}}\phi + \frac{H_4}{\sqrt{\frac{H_1\rho_{g_0}}{n_2 P}}}\phi^2, \tag{36}$$

with $H_4$,

$$H_4 = -\frac{1}{2}\frac{H_1\rho_{h_0}{}^2 n_1 - 4H_2 n_1\rho_{h_0}\rho_{g_0} + 2H_2 n_2\rho_{h_0}\rho_{g_0} + 3H_3 n_1\rho_{g_0}{}^2 - 2H_3 n_2\rho_{g_0}{}^2}{H_1 n_1\rho_{h_0}{}^2}$$
$$+\frac{3}{8}\frac{(H_1 n_1\rho_{h_0} - 2H_2 n_1\rho_{g_0} + H_2 n_2\rho_{g_0})^2}{H_1{}^2 n_1{}^2\rho_{h_0}{}^2}, \tag{37}$$

and $L_1$, $L_2$, $L_3$ and $H_1$, $H_2$, $H_3$ are defined as,

$$L_1 = \left(\frac{P_0}{P}\right)^{\frac{1}{n_1}}, \; L_2 = \left(\frac{P_0}{P}\right)^{\frac{1}{n_2}}, \; L_3 = \frac{L_1}{\rho_{h_0}} - \frac{L_2}{\rho_{g_0}},$$

$$H_1 = \left(\frac{P_0}{P}\right)^{-\frac{1}{n_2}}, \; H_2 = \left(\frac{P_0}{P}\right)^{-\frac{2n_1-n_2}{n_1 n_2}}, \; H_3 = \left(\frac{P_0}{P}\right)^{-\frac{3n_1-2n_2}{n_1 n_2}}. \tag{38}$$

To find the optimal values of $\rho(\phi)$ and $c(\phi)$ with respect to the parameter mass ratio $\phi$, we look for the zeros of the gradient of the functions $\rho(\phi)$ and $c(\phi)$ as follows (see reference [24]),

$$\frac{\partial\rho}{\partial\phi} = -\frac{\rho_{g_0}{}^2}{L_2{}^2}\left(\frac{L_1}{\rho_{h_0}} - \frac{L_2}{\rho_{g_0}}\right) - 2\frac{\rho_{g_0}{}^2(L_1\rho_{g_0} + L_2\rho_{h_0})\phi}{L_2{}^3\rho_{h_0}}\left(\frac{L_1}{\rho_{h_0}} - \frac{L_2}{\rho_{g_0}}\right), \tag{39}$$

$$\frac{\partial c}{\partial\phi} = -\frac{1}{2}\frac{H_1 n_1\rho_{h_0} - 2H_2 n_1\rho_{g_0} + H_2 n_2\rho_{g_0}}{H_1 n_1\rho_{h_0}} \cdot \frac{1}{\sqrt{\frac{H_2\rho_{g_0}}{n_2 P}}} + 2\phi\frac{H_5}{\sqrt{\frac{H_1\rho_{g_0}}{n_2 P}}}, \tag{40}$$

with $H_5$ defined as follows,

$$H_5 = -\frac{1}{2}\frac{H_1 n_1\rho_{h_0}{}^2 - 4H_2 n_1\rho_{g_0}\rho_{h_0} + 2H_2 n_2\rho_{h_0}\rho_{g_0} + 3H_3 n_1\rho_{g_0}{}^2 - 2H_3 n_2\rho_{g_0}{}^2}{H_1{}^2 n_1{}^2\rho_{h_0}{}^2}$$
$$+\frac{3}{8}\frac{(H_1 n_1\rho_{h_0} - 2H_2 n_1\rho_{g_0} + H_2 n_2\rho_{g_0})^2}{H_1{}^2 n_1{}^2\rho_{h_0}{}^2}, \tag{41}$$

and by solving the equations $\frac{\partial\rho}{\partial\phi} = 0$ and $\frac{\partial c}{\partial\phi} = 0$ we can obtain the optimal values of the mass ratio $\phi$ explicitly as follows,

$$\phi_{\text{optimal},\rho} = \frac{1}{2} \cdot \frac{L_2\rho_{h_0}}{L_1\rho_{g_0} - L_2\rho_{h_0}}, \tag{42}$$

$$\phi_{\text{optimal},c} = \frac{-2[H_1 n_1\rho_{h_0} - 2H_2 n_1\rho_{g_0} + H_2 n_2\rho_{g_0}]H_1 n_1\rho_{h_0}}{H_6}, \tag{43}$$

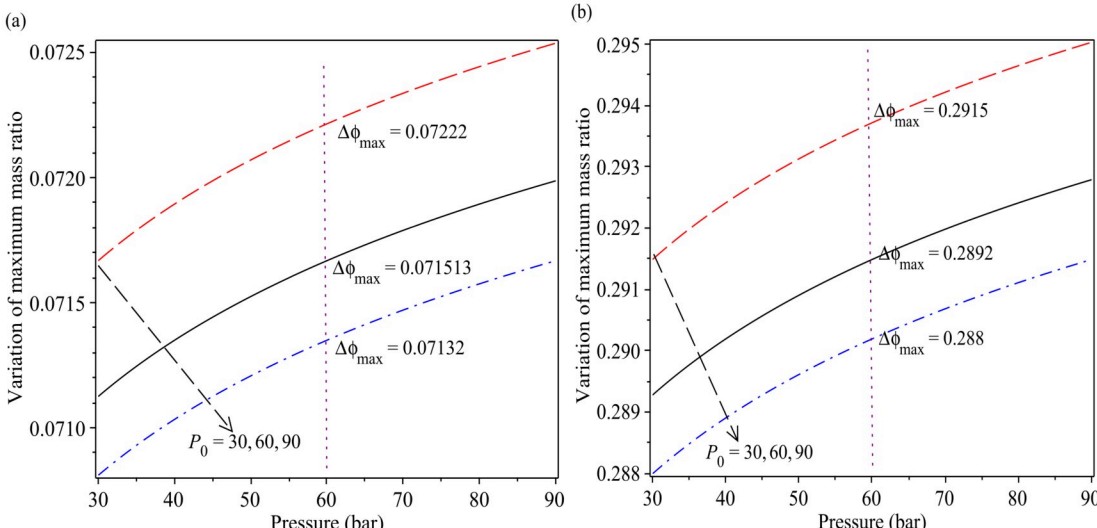

**Fig 9. Variation of optimal mass ratio as a function of pressure with different initial pressure.** (a): using Eq (8) and (b): using Eq (10).

with $H_6$ defined as follows,

$$H_6 = H_1{}^2 n_1{}^2 \rho_{h_0}{}^2 - 4H_1 H_2 n_1{}^2 \rho_{h_0} \rho_{g_0} + 2H_1 H_2 n_1 n_2 \rho_{h_0} \rho_{g_0} + 12H_1 H_3 n_1{}^2 \rho_{g_0}{}^2 - 8H_1 H_3 n_1 n_2 \rho_{g_0}{}^2 - 12H_2{}^2 n_1{}^2 \rho_{g_0}{}^2 + 12H_2{}^2 n_1 n_2 \rho_{g_0}{}^2 - 3H_2{}^2 n_2{}^2 \rho_{g_0}{}^2. \tag{44}$$

Fig 9 presents the variation of optimal mass ratio $\phi$ as a function of the pressures $P$ and $P_0$ by using the Eqs (35) and (36). Fig 9a and 9b show that the maximum and minimum values for the optimum mass ratio variations respectively are as, $\min_\rho \phi_{\text{optimal}} = 0.0708$, $\max_\rho \phi_{\text{optimal}} = 0.0725$ and $\min_c \phi_{\text{optimal}} = 0.288$, $\max_c \phi_{\text{optimal}} = 0.295$. As seen from the results with increasing the initial pressure $P_0$ from 30 up to 90, the variation optimum mass ratio increases. For $P = 60$, and different values of initial pressure $P_0 = 30, 60, 90$ we have the max mass ratio variations as $\Delta\phi_{\text{optimal},\rho} = 0.07222, 0.07113, 0.07132$ $\Delta\phi_{\text{optimal},c} = 0.2915, 0.2892, 0.2882$. This means that for controlling the optimal pressure we must decrease $\nabla\phi$ or increase $\nabla P_0$ (see Fig 10).

Fig 10 shows the transient pressure of hydrogen natural gas mixture for isothermal flow when leakage occurs at $X_L = L/3$ in horizontal pipeline; red is transient pressure for gas mixture $\phi = 0.25$ and black is transient pressure for gas mixture $\phi = 0.5$. As seen from this figure for $\nabla\phi = 0.25 - 0 = 0.25$ we have $\nabla P = 8.708$ and $\nabla\phi = 0.5 - 0 = 0.5$ we have $\nabla P = 10.031$ ($P$ is transient pressure).

## Conclusion

The Taylor series approximation coupled with regression analysis was used to solve the flow equations of hydrogen natural gas mixture in an inclined pipeline. To validate the approximation series, Fig 3 shows a comparison of the density evolution with pressure between exact solution and approximation series, for different values of the hydrogen mass fraction $\phi$, by assuming an initial pressure $P_0 = 35$ bar and $T_0 = 15\,°C = 288$ K. Fig 4 shows a validation between the regression polynomials of pressure $P(x)$ (Eq (23)) with the numerical results of Elaoud et al. and Subani et al. papers for different values of mass ratio $\phi = 0, 0.5, 1$. The results

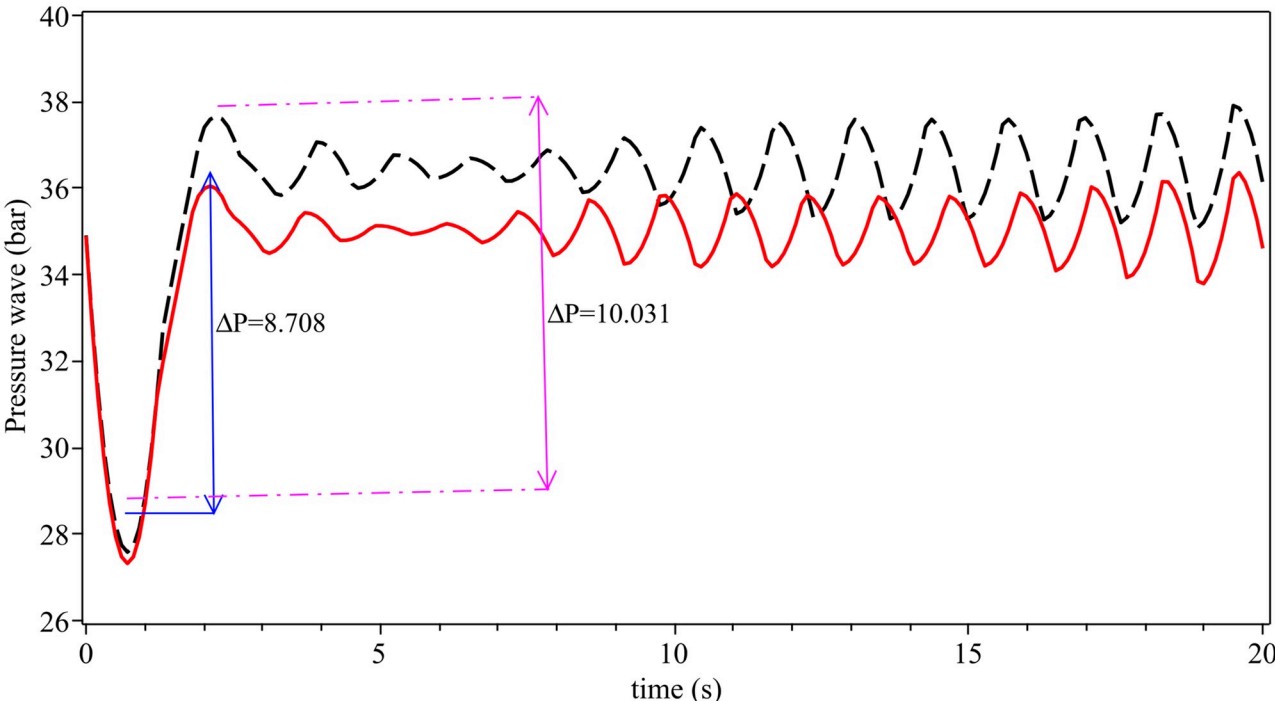

**Fig 10. Transient pressure of hydrogen natural gas mixture for isothermal flow when leakage occurs at $X_L = L/3$ in horizontal pipeline.** Red solid line: $\phi = 0.25$ and black dash line: $\phi = 0.5$.

in Figs 5 and 6 show that the obtained results using proposed method are in good agreement with those of reduced order modelling (ROM) and the method of characteristics (MOC). Then, our method is working as well as other methods and give the smoother results.

A method called Zero Gradient Control (ZGC) was applied to control the optimal transient pressure. The basic concept ZGC is to develop algebraic terms of density and celerity pressure wave equations for the partial derivatives with respect to initial pressure, mass ratio and transient pressure. The great advantage of ZGC is to not to have to solve a nonlinear system of equations which would allow to determine equations for the optimal set points that are not coupled. The equations derived for density and celerity pressure wave can be applied to many systems in different fields. According to the results of the proposed method the ratio of pressure with respect to the mass ratio $\phi$ and initial pressure $P_0$ are 1.22 and 1.011, respectively (see Figs 7 and 8). The results show that the mass ratio is important for controlling the optimal pressure in an inclined pipeline. Fig 9 shows that for $P = 60$, and different values of initial pressure $P_0 = 30, 60, 90$ we have the optimal mass ratio variations as $\Delta\phi_{optimal,\rho} = 0.07222, 0.07113, 0.07132$ and $\Delta\phi_{optimal,c} = 0.2915, 0.2892, 0.2882$. This means that for controlling the optimal pressure we must decrease $\nabla\phi$ or increase $\nabla P_0$ (see Fig 10). As seen from Fig 10 for $\nabla\phi = 0.25 - 0 = 0.25$ we have $\nabla P = 8.708$ and $\nabla\phi = 0.5 - 0 = 0.5$ we have $\nabla P = 10.031$.

## Supporting information

**S1 Fig. Graphical abstract for mathematical analysis of gas mixture pressure control.**
(PDF)

**S1 Program. To approximate Taylor series for density and celerity.**
(PDF)

**S2 Program. To find the regression polynomials for pressureand velocity.**
(PDF)

**S3 Program. To solve Eqs (2) and (3).**
(PDF)

**S4 Program. Zero gradient control for controlling the pressure.**
(PDF)

**S5 Program. Pressure control by reining mass ratio.**
(PDF)

## Acknowledgments

Financial support provided by Research and Innovation University Grant Scheme, Universiti Teknologi Malaysia, is gratefully acknowledged.

## Author Contributions

**Funding acquisition:** Norsarahaida Amin.

**Methodology:** Sarkhosh S. Chaharborj.

**Software:** Sarkhosh S. Chaharborj.

**Supervision:** Norsarahaida Amin.

**Writing – original draft:** Sarkhosh S. Chaharborj.

**Writing – review & editing:** Norsarahaida Amin.

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
