## [Decision Letter · Decision Letter 0]

27 Nov 2019

PONE-D-19-28982

Controlling optimal pressure of hydrogen-natural gas mixture in an inclined pipeline

PLOS ONE

Dear Dr. Chaharborj,

Thank you for submitting your manuscript to PLOS ONE. After careful consideration, we feel that it has merit but does not fully meet PLOS ONE’s publication criteria as it currently stands. Therefore, we invite you to submit a revised version of the manuscript that addresses the points raised during the review process.

Please revised the manuscript by considering the reviewers' comments.

We would appreciate receiving your revised manuscript by Jan 11 2020 11:59PM. To enhance the reproducibility of your results, we recommend that if applicable you deposit your laboratory protocols in protocols.io, where a protocol can be assigned its own identifier (DOI) such that it can be cited independently in the future. For instructions see: http://journals.plos.org/plosone/s/submission-guidelines#loc-laboratory-protocols

We look forward to receiving your revised manuscript.

Kind regards,

Jie Zhang

Academic Editor

PLOS ONE

Journal Requirements:

Additional Editor Comments (if provided):

Reviewers' comments:

Reviewer's Responses to Questions

**Comments to the Author**

1. Is the manuscript technically sound, and do the data support the conclusions?

Reviewer #1: Yes

Reviewer #2: Partly

2. Has the statistical analysis been performed appropriately and rigorously? 

Reviewer #1: Yes

Reviewer #2: I Don't Know

3. Have the authors made all data underlying the findings in their manuscript fully available?

Reviewer #1: Yes

Reviewer #2: Yes

4. Is the manuscript presented in an intelligible fashion and written in standard English?

Reviewer #1: Yes

Reviewer #2: No

5. Review Comments to the Author

Reviewer #1: 1. Turnitin results showed a high similarity index (29%) for this paper (exclude quotes on, exclude bibliography on and exclude matches < 1%). Please revise first.

2. A method called Zero Gradient Control (ZGC) was said to control the optimal transient pressure. However, it is not clear the performance of the transient pressure for regulatory or servo problem.

3. Please state clearly whether the optimal set point is for controller equipment such as PID controller, or else with different meaning.

Reviewer #2: Please put more effort in correct English. Furthermore, variables are not named uniform and often units are missing etc. In general you have written a nice piece of work, but it cannot be published yet in its current form. See attachment for further comments

6. PLOS authors have the option to publish the peer review history of their article (what does this mean?). If published, this will include your full peer review and any attached files.

Reviewer #1: No

Reviewer #2: No

---

## [Author Response · Author response to Decision Letter 0]

14 Jan 2020

We warmly thank the very competent Reviewers for the time spent on reading the manuscript and for the valuable comments and suggestions. We have modified the manuscript accordingly, and the detailed corrections are listed point by point in the attached file "Response to Reviewers".

Please see the attached file "Response to Reviewers" for reviewer and editor comments. 

Thank you very much

---

## [Decision Letter · Decision Letter 1]

28 Jan 2020

Controlling the pressure of hydrogen-natural gas mixture in an inclined pipeline

PONE-D-19-28982R1

Dear Dr. Chaharborj,

We are pleased to inform you that your manuscript has been judged scientifically suitable for publication and will be formally accepted for publication once it complies with all outstanding technical requirements.

With kind regards,

Jie Zhang

Academic Editor

PLOS ONE

Additional Editor Comments (optional):

Reviewers' comments:

Reviewer's Responses to Questions

**Comments to the Author**

1. If the authors have adequately addressed your comments raised in a previous round of review and you feel that this manuscript is now acceptable for publication, you may indicate that here to bypass the “Comments to the Author” section, enter your conflict of interest statement in the “Confidential to Editor” section, and submit your "Accept" recommendation.

Reviewer #1: (No Response)

2. Is the manuscript technically sound, and do the data support the conclusions?

Reviewer #1: Yes

3. Has the statistical analysis been performed appropriately and rigorously? 

Reviewer #1: Yes

4. Have the authors made all data underlying the findings in their manuscript fully available?

Reviewer #1: Yes

5. Is the manuscript presented in an intelligible fashion and written in standard English?

Reviewer #1: Yes

6. Review Comments to the Author

Reviewer #1: (No Response)

7. PLOS authors have the option to publish the peer review history of their article (what does this mean?). If published, this will include your full peer review and any attached files.

Reviewer #1: No

---

## [Editor Report · Acceptance letter]

18 Feb 2020

PONE-D-19-28982R1 

Controlling the pressure of hydrogen-natural gas mixture in an inclined pipeline 

Dear Dr. Chaharborj:

I am pleased to inform you that your manuscript has been deemed suitable for publication in PLOS ONE. Congratulations! Your manuscript is now with our production department. 

With kind regards,

on behalf of

Dr. Jie Zhang 

Academic Editor

PLOS ONE